# Value Analysis Model to Support the Building Design Process



**Zulay Giménez [1]**, **Claudio Mourgues [1]**, **Luis F. Alarcón [1]**, **Harrison Mesa [2]** and **Eugenio Pellicer [3,*]**

[1] School of Engineering, Department of Construction Engineering and Management, Pontificia Universidad Católica de Chile, Avda. Vicuña Mackenna 4860, Macul, Santiago, Chile; zmgimenez@ing.puc.cl (Z.G.); cmourgue@ing.puc.cl (C.M.); lalarcon@ing.puc.cl (L.F.A.)

[2] School of Civil Construction, Pontificia Universidad Católica de Chile, Avda. Vicuña Mackenna 4860, Macul, Santiago, Chile; hmesa@uc.cl

[3] School of Civil Engineering, Universitat Politècnica de València, Camino de Vera s/n, 46022 Valencia, Spain

* Correspondence: pellicer@upv.es

**Abstract:** The architecture, engineering, and construction industry requires methods that link the capture of customer requirements with the continuous measurement of the value generated and the identification of value losses in the design process. A value analysis model (VAM) is proposed to measure the value creation expected by customers and to identify value losses through indexes. As points of reference, the model takes the Kano model and target costing, which is used in the building project design process. The VAM was developed under the design science research methodology, which focuses on solving practical problems by producing outputs by iteration. The resulting VAM allowed the measurement and analysis of value through desired, potential, and generated value indexes, value loss identification, and percentages of value fulfillment concerning the design stage. The VAM permits the comparison of different projects, visualization of the evolution of value generation, and identification of value losses to be eradicated. The VAM encourages constant feedback and has potential to deliver higher value, as it enables the determination of parameters that add value for different stakeholders and informs designers where to direct resources and efforts to enhance vital variables and not trivial variables.

**Keywords:** value generation; value loss; desired value; potential value; value indexes; design science research

## 1. Introduction

The design process in the architecture, engineering, and construction (AEC) industry is unable to respond to the value creation expectations of the customer [1], nor does it use rigorous methods that measure value or identify and control value losses [2,3]. A design initially requires the establishment of customer requirements, which are usually incomplete, poorly formulated, and ambiguous [4], to generate a complete design and to then evaluate aspects of cost, time, quality, and other criteria. These aspects, when considered late, do not necessarily correspond to the clients' value requirements [5], and historically have been exceeded or deviated from [6], producing consequences of inefficiency and lack of quality and productivity in projects [7]. Satisfying clients involves understanding and resolving their different perspectives and restating their needs in construction terms [8].

Design is an interactive and multidimensional effort that should represent the interests of several stakeholders and customers [9]. However, the inability to study, understand, and consider customer needs within the industry is widely recognized [10], as even customer interaction in the design process

is perceived as a nuisance [11]. Womack and Jones [12] consider goods and services that do not respond to user needs as waste within design. If customer value is not fully understood in a project, the project is very likely to result in low compliance with customer expectations or multiple modifications during the project [13].

To date, it is not possible to measure the value delivered to project customers, not only in regard to costs or objective measurements, but also concerning compliance with requirements and the evolution of the value perceived by customers over time [14]. In addition, it is expected that some customer requirements may be lost during design [15], but these value losses are generally not discovered in the process [16] or are identified late in the construction stage [17].

Considering this gap in the body of knowledge, the purpose of this paper is to respond to the need to measure the value creation expected by different customers within the design process through indicators, and to contribute to the early identification of value losses to control them in time. A value analysis model (VAM) is proposed to measure the value creation expected by customers and to identify value losses in the building project design process through indexes that take the Kano model [18] and target costing [19] as points of reference. VAM was developed under the design science research (DSR) methodology, which focuses on solving practical problems and producing artifacts as outputs [20]. One of the main contributions of the model proposed in this study is the possibility of better understanding the concept of value and how to capture and measure it, as well as knowing when and how value can be lost to support the conditions of customer satisfaction.

## 2. Research Method

### 2.1. Overall Approach

The VAM was developed on the conceptual basis of design science research (DSR). DSR is used to explore new solution alternatives to solve problems and to develop or create an artifact [20]. Such artifacts are potentially constructs, models, methods, or any designed object in which a research contribution is incorporated into the design [21]. DSR bridges the gaps among the contextual environment of the research project, design science activities, and the knowledge base of scientific foundations, experience, and expertise, iterating between the activities of construction and evaluation of research design artifacts and processes [22].

Figure 1 presents the research approach based on the DSR process model proposed by Peffers et al. [21], which comprises five iterative steps: Problem identification and motivation; definition of the objectives; design and development; demonstration; and evaluation. In this case, the developed artifact is the VAM. Regarding problem identification, a literature review is performed on value and customer concepts, generation and loss of value, and value-related methods in the AEC industry. The VAM artifact was developed in two complete iterations using different case studies in each cycle to deliver three drafts. The first cycle used a project present in the literature as a case study: An application of the Kano model in requirements analysis of a company's consulting project located in Guangzhou [23]. The second cycle utilized two projects of a Chilean real estate and construction company, whose primary activity is the integral execution of high-rise residential buildings. Each cycle included the five steps introduced previously: Identification, definition, design and development, demonstration, and evaluation. The following sub-sections explain in-depth each one of these activities within their corresponding iterations.

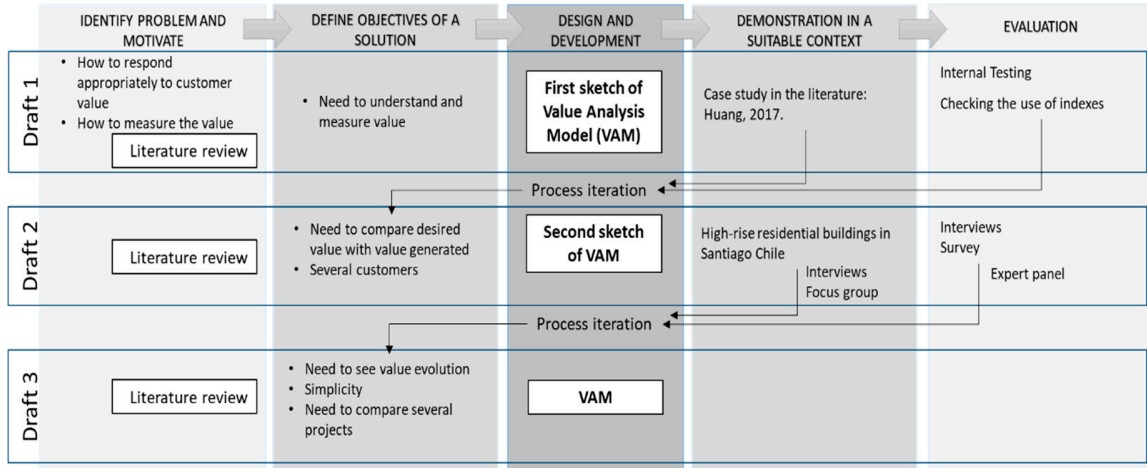

**Figure 1.** Research approach based on design science research (DSR).

*2.2. Problem Identification and Motivation: Literature Review*

2.2.1. Value

Different value concepts with similar approaches have been presented in the literature. Value is generally expressed as a relationship between two aspects, such as function and the total life cycle cost of that function [24] or costs and benefits [25]. Other authors express value as the relationship between the effectiveness of a product in achieving the objectives and the resources consumed, what you get and what you give, or the balance between the benefits and sacrifices involved in value judgments [26–28]. All these definitions can be summarized as the relationship between the satisfaction of needs and the use of required resources.

Value can be seen in several ways by different customers in diverse situations. Value will be defined differently by each stakeholder depending on his or her judgment of the factors given and received, just as value depends on the theoretical context and on subjective perceptions and evaluative judgments [29]. That is, what is value for one may not have any value for others [3]. On the other hand, value may vary over time [30], as customers' needs are dynamic [31] and the context may change; therefore, value judgments made at different times will differ [29]. The value generated through the projects and activities is not static but flows (ripple effect) to generate value in other areas in the present and in the future to benefit different stakeholders. [30]. In this value dynamism, one can distinguish the pre-use value, also called the expected or desired value, and the post-use value, also called received or perceived value [32–34]. In the context of this research, value is defined as the fulfillment of the needs of different customers considering their diverse visions, the dynamism of value over time, and the resources contributing to value generation.

2.2.2. Customer

An essential consideration for value management is the impact of the customer on the project process [27]. In business terminology, the words "customer" (product buyer) and "consumer" (end-user of the product) are often used interchangeably [31]. However, the customer may be different from the end-user [35]. In quality management, the customer concept is broadened by considering external customers (any person who is not part of a company and purchases its products and/or services) and internal customers (any person who is part of a company and who receives a product—information, materials, or parts—to which he or she adds his or her own work and delivers it to another customer) [36]. According to Kamara [37], the client should represent the interests of users and other identified persons, groups, or organizations who influence and/or are affected by the acquisition, use, operation, and demolition of the facility being commissioned. In a similar sense,

Drevland and Tillmann [38] relate the customer to all the people who are somehow affected by a project (stakeholders), and these authors classify the customer and stakeholders within a single group because of the relationship between them. In the context of this research, the term "customer" will be used interchangeably with "client" and "stakeholder."

### 2.2.3. Generation and Loss of Value

The process of generating value has been discussed from many points of view. Leinonen and Huovila [39] define this process in three phases: (1) Determining the customer's requirements, (2) creating solutions to meet these requirements, and (3) verifying during the project that these requirements are met in the best way possible. Customer requirements refer to the objectives, needs, wishes, and expectations of the customer. These requirements should be a description of the functions, attributes, or other special features of the facility necessary to satisfy the needs of the customer [37]. Zhang et al. [40] relate value generation to maximizing value, minimizing the life cycle cost, and considering customer needs. Value maximization can be achieved by balancing the number of needs met with the resources used. Koskela [3], on the other hand, defines five principles of value generation within the production process, relating them to the internal functions of the supplier and the customer:

- Requirements capture: Ensuring that all customer requirements, both explicit and implicit, have been captured as the first step in generating value.
- Requirements flow-down: Ensuring that all relevant customer requirements are retained in all phases of production and are not lost when progressively transformed into design solutions, production plans, and products.
- Comprehensive requirements: Ensuring that all requirements relate to all customer roles.
- Production subsystem capacity: Ensuring the capacity of the production system to produce products as needed.
- Value measurement: Through metrics, ensuring that value is generated for the customer.

Additionally, Koskela [3] incorporates the term loss of value to refer to the part of value that is not provided, even if providing it is potentially possible. This concept is a way of measuring value in relative terms, that is, the value achieved compared with the best possible value. For their part, Womack and Jones [12] suggest considering the provision of an incorrect product or service as waste. From the perspective of value, waste is the loss of value, defined by a situation in which a product is not used correctly, there is a loss of quality, tasks are not performed in the way they should be, or byproducts with harmful or undesirable value are obtained [16]. One way of determining the notion of "potentially possible" is to look at competitors; if they provide more value, providing more value is also potentially possible for the company in question. Another way is to estimate the value when the whole cycle of product realization is ideal [3].

Integrating these perspectives, the following elements are summarized below as influential factors within the value generation process:

- Minimization of the life cycle cost.
- Pursuit of the satisfaction of customers' needs.
- Pursuit of value maximization.
- Requirements capture.
- Requirements flow-down.
- Pursuit of integrated solutions for the fulfillment of requirements.
- Assurance of the capacity and performance of the production system.
- Verification that the requirements are met.
- Value measurement through metrics.
- Identification of value losses.

### 2.2.4. Value-Related Methods in the Architecture, Engineering, and Construction (AEC) Industry

Value management, also known as value analysis (VA), value methodology, or value engineering [25], is a management style that has evolved from previous methods based on the concept of value and the functional approach. These methodologies were first proposed in the 1940s and 1905s by Lawrence D. Miles, who developed the VA technique as a method for improving the value of existing products [24]. Initially, VA was used to identify and eliminate unnecessary costs. However, it is equally effective in increasing performance and addressing non-cost resources [26].

In general, value is understood in terms of cost, price, or monetary aspects [28]. However, others focus on customer voice and preferences, such as stated preferences [33], evidence-based design [41], the design performance measurement matrix [42], design thinking [43], the design value scorecard [44], agile transform development [45], the value chain model [46], the balanced scorecard [27], the maximum difference method or Best-Worst approach [47], the voice of the customer (VOC) [48], and the framework for value-optimized design [49].

Within the context of the AEC industry, methods such as post-occupancy evaluation (POE) [50], virtual design and construction (VDC) [51], and target value design (TVD) [52] have been created and used. Value-related methods used in other industries have also been incorporated into the AEC industry, such as stated preferences, design thinking, value engineering [25], quality function deployment (QFD) [31], the Kano model [53], and target costing [54]. Furthermore, additional methods have been developed for some "ad hoc" needs within the AEC industry [1,55–57].

Table 1 summarizes information on the methods used in the AEC industry and their relationship with the influential factors within the value generation process. The factors established in 2.2.3 were considered, including the relationship with other factors such as quality, constructability, and productivity; although they may be part of a customer's requirements, they are often confused with the definition of value [58].

In one way or another, all methods aim to satisfy customer requirements. However, to comply with them, methods do not necessarily focus on their capture, flow, and subsequent verification of compliance; rather, they focus on aspects of quality, constructability, and productivity. The limited use of strategies to capture requirements or to identify value losses during the design process, the nonconsideration of the assurance of the production system's capacity, and the generalized lack of the use of metrics or indexes related to value are also visualized.

**Table 1.** Methods and their relationship with value generation.

| Methods | Minimization of the Life Cycle Cost | Pursuit of Satisfaction of the Customer's Needs | Pursuit of Value Maximization | Requirements Capture | Requirements Flow-Down | Pursuit of Integral Solutions for the Fulfillment of Requirements | Assurance of the Capacity and Performance of the Production System | Verification That the Requirements Are Met | Value Measurement through Metrics | Identification of Value Losses | Relationship with Quality, Constructability, Productivity | Reference |
|---|---|---|---|---|---|---|---|---|---|---|---|---|
| Post-occupancy evaluation (POE) | | X | | | | | | X | | X | X | [27,50] |
| Value Management/ Engineering | X | X | | | | | | | | | X | [25,59] |
| Kano Model | | X | | X | | | | | | | X | [23,60,61] |
| Quality Function Deployment (QFD) | | X | | X | X | X | | X | | | X | [31,62,63] |
| Target Costing | X | X | X | | | | | X | | | X | [54,64] |
| Virtual Design and Construction (VDC) | X | X | X | | X | X | X | | | X | X | [5,65–67] |
| Target Value Design (TVD) | X | X | X | | X | X | | X | | | X | [41,52] |
| Assessment of Housing Projects | X | X | X | | | | | | X | | X | [55] |
| 3Cv + 2 | | X | X | | | | | X | X | | X | [56] |
| Framework for Enhancing Value Creation in Construction Projects | | X | | | X | | | | | | | [57] |
| Owner Value Interest Model | | X | X | X | | | | | | | X | [1] |
| Stated Preferences | | X | | X | | | | | X | | | [33] |
| Design Thinking | | X | | | X | X | | X | | | X | [43] |

Regarding requirements capture, only four models have this emphasis: Kano, QFD, owner value interest, and stated preferences. QFD considers the capture of requirements only as a list of customer

wishes, without considering any order of importance [62]. The owner value interest model and the stated preferences model evaluate the degree of importance of each attribute or characteristic of value [1,33]. The Kano model measures customer feelings and the impact of product/service quality on customers' perceived satisfaction, classifying attributes according to their influence on customer satisfaction [23].

Concerning the identification of value losses, POE and VDC are discussed. POE is an evaluation of an inhabited property after use by a user; thus, value losses are identified too late to be corrected in time. Regarding VDC, its main contribution is the possibility of building virtually as the design is developed, thus achieving in a timely manner the identification of inconsistencies between design disciplines, aspects of quality and constructability, value loss in the design process itself, and the designed product. VDC is the use of integrated multidisciplinary performance models of design and construction projects to support business objectives, and it is used to emphasize product, organizational, and processual aspects [68]. It is a value-related method in the AEC industry that considers ensuring the capacity of the production system.

Regarding the use of metrics or indexes related to value, Pandolfo et al. [55] establish metrics of the importance perceived by customers of specific attributes with regard to their percentage within the cost to balance the use of resources with the "value" of the attribute. García et al. [56] focus on quality in the construction phase and beyond. Stated preferences evaluate the degree of importance of each attribute or characteristic and then determine a specific variable, called the general significance index (GSI).

None of the methods have all the factors considered influential or present in the generation of value. For this reason, the AEC industry has used them together to balance those that are missing. Among the methods that have comprehensive approaches to the most significant number of factors, VDC and TVD stand out. However, these two methods are notorious because they do not capture requirements in a systematic way or measure value through metrics or indexes.

### 2.2.5. Point of Departure

This literature review highlights a gap in current practices regarding the value generation of the design process within the AEC industry: There is a lack of adequate methods that link the suitable capture of customer requirements with the continuous measurement of the value generated as well as the timely identification of value losses at the time of design and not later, when it is no longer feasible to deal with them. There is a lack of indexes that allow value to be measured in an integral way considering the different perspectives of customers. The proposed value analysis model (VAM) can help designers and project managers improve decision making within the design process, increase customer satisfaction, and evaluate the allocation of resources to activities that generate value.

### 2.3. Draft 1

The first draft of the VAM addresses the need to understand and measure value in the design process. As points of reference, VAM takes the attractive quality theory of Kano et al. [18], also known as the Kano model, as well as the coefficient of satisfaction (CS) of Berger et al. [60]. The Kano attribute classification allows requirements to be assessed according to the perception of the customer to calculate the desired value and the potential value of the process, by-products, and products of the design.

Kano et al. [18] fundamentally distinguish the following types of attributes [53,69]: (1) Must-be attributes (M), which are essential elements of a product that contribute only to avoiding dissatisfaction; (2) one-dimensional attributes (O), in which customer satisfaction is proportional to the level of compliance with these attributes; (3) attractive attributes (A), which are attributes that have a significant influence on customer satisfaction because they meet the tacit needs and not just the explicit needs of the customer; (4) indifferent attributes (I), which are attributes that do not play a role in determining

customer satisfaction; and (5) reverse attributes (R), which are product characteristics that are not only undesirable but also the opposite of what is expected.

Additionally, Kano et al. [23] incorporate a requirements capture instrument that overcomes the bias that arises from traditional requirement survey instruments. Their instrument uses a two-dimensional questionnaire for each attribute to classify them. The first question is functional or positive (how do customers feel if the proposed characteristic is provided?); and the second question is dysfunctional or negative (how do customers feel if the intended characteristic is not provided?).

Kano classifies each requirement according to most of the answers, which would not be statistically correct because, in general, the answers tend to be dispersed in several categories. For this reason, Berger et al. [60] incorporate the CS, which is composed of two indexes (satisfaction—SI and dissatisfaction—DI) that represent, respectively, a positive number or the relative value of compliance with this customer requirement and a negative number or the relative cost of not meeting this customer requirement (see Equations (1) and (2)). The CS positions each of the attributes in four possible quadrants: A, I, M, and O, thus contributing to the appropriate classification of the attributes according to Kano.

$$SI = (O + A) / (M + O + A + I) \qquad (1)$$

$$DI = (M + O) / (M + O + A + I) \qquad (2)$$

In this first draft, it creates desired value and potential value indexes that represent the minimum and maximum value, respectively, needed to achieve the customer's requirements. A case study from the literature [23] was used to test the first draft of the VAM, which applies the Kano model to analyze the requirements of a project consulting firm based in Guangzhou whose main activity is the design and construction of roads. Huang [23] establishes 18 attributes and classifies them using the Kano model, administering the two-dimensional questionnaire to 41 professionals among the company's managers and staff.

The results of this case study were applied to test the calculation of the value indexes and the relationships between them, the inclusion of the reverse attributes within the satisfaction coefficient, as well as different hypothetical scenarios of value generation and loss. After the use of VAM in this case study, the desired and potential value must be compared with the value generated, in addition to measuring value for the different customers present in the design process, such as owner, users, designers, and builders.

*2.4. Draft 2*

In Draft 2 of the model, the value indexes were generated, and the compliance and loss of value percentages were included to address the needs arising after testing the first draft. In addition, the second draft of the model identifies the relationship between the types of attributes and the generation or not of value.

The model was tested in two projects in the preliminary stages of the design process involving a real estate and construction company located in Santiago, Chile, whose main activity is the integral execution of high-rise residential buildings. VAM was applied in focus group meetings consisting of a cluster of 20 professionals that included directors and professionals from the company, such as architects, civil engineers, industrial engineers, and architectural engineers.

Initially, the focus group identified six main groups of customers. Later, it established for each type of customer a percentage according to the level of importance of each one. The different percentages were used as a weighting factor (W): Users (30.8%), owners (20.8%), designers (14.3%), builders (15.8%), reviewers (7.5%), and suppliers (10.8%). However, the state of progress (preliminary design) of both projects did not allow the incorporation of users, reviewers, and suppliers. For this reason, in this case, the VAM was applied only to designers, builders, and owners. The weighting factors were again established as follows: Owners (42%), designers (28%), and builders (31%).

In order to create the lists of attributes, interviews were conducted with the professionals about the positive attributes to be accepted and the negative ones to be avoided in both the design process and the product. Two types of products were established: A physical design product or deliverables and a conceptual or potentially buildable product after the design process. The final value attributes were identified and refined through an iterative review and revision process.

Subsequently, two-dimensional questionnaires were administered to owners, designers, and builders, such as surveys, to collect the first results regarding the desired value and the potential value of the process and the design products. Finally, the value generated in both projects thus far was measured.

Additionally, an expert panel was conducted through individual interviews in which the characteristics of the VAM and its operation were presented. This validation was achieved through an academic-industrial specialist panel, as shown in Table 2. Consistency, connection, coherency, simplicity, completeness, theoretically-based association, exactness, clarity, and use logic were checked.

In addition, the functional structure of the VAM was presented by Giménez et al. [70] at an international event with experts in value generation and lean management issues. After this interaction with experts (panel and congress), the need emerged to compare the evolution value over time and increase the simplicity in showing and applying the model.

**Table 2.** Expert panel.

| Profession | Occupation | Experience |
| --- | --- | --- |
| Ph.D. Civil Engineer | Senior professor at a public university in Venezuela | 33 years of experience in construction management and quality |
| Ph.D. Civil Engineer | Senior professor at a public university in Spain | 28 years of experience in construction management |
| Ph.D. Civil Engineer | Project Manager. Bogotá, Colombia | 10 years of experience in construction |
| Ph.D. Civil Engineer | Corporate quality leader. USA | 13 years of experience in lean design and construction |
| MSc. Architect | Leader in Lean design and integrated project delivery (IPD). USA | 25 years of experience in lean design |
| Ph.D. Civil Engineer | Associate professor at a public university in Brazil | 30 years of experience in construction management and economics |
| Civil Engineer | Talent development manager/LCI instructor. Perú | 14 years of experience in lean design and construction |
| MSc. Civil Engineer | Lean consultant. USA | 34 years of experience in design and construction |

*2.5. Draft 3*

For version 3 of the VAM, the possibility of several revisions of the value generated throughout the design project was incorporated, and the format of the questionnaires was simplified to meet the needs that emerged after evaluating the second draft. In response to the literature review, experiences related to the target cost [19] were included. Target costing is a disciplined process of determining the total cost of making a proposed product with specific functionality to generate the desired profitability at its selling price [64]. In target costing, product design costs increase continuously until the allowable cost and the target cost [41], which represent the willingness to pay and the customer's requirements and competitive conditions, respectively [54], are reached.

Like target costing, which iterates the design to achieve the allowable cost and target cost, the VAM has as its highest goal to accomplish in the design iterations the potential value, but if the desired value is achieved, the project is "valuably" feasible. In addition, the ease with which the model can be applied in different contexts has been improved. This paper introduces the latest version of this VAM.

## 3. Results

*3.1. Value Analysis Model (VAM)—General Overview*

Next, a general overview of the VAM corresponding to Draft 3 of the model is presented. Each customer has requirements that represent design inputs. The preliminary stage of the design includes requirements capture, in which the customer's expectations regarding the product and the design process are captured, represented by the desired value (DV) and the potential value (PV). As a result of this first stage, the desired value and potential value indexes (DVI; PVI) of the process, the product, or both are obtained. Next, the design process begins; in this stage, the value that should respond to the desired value and could respond to the potential value is generated. As a result of this second stage, the generated value indexes (GVI) are obtained, which differ in the desired value generated (DVG) and potential value generated (PVG) of the process, the product, or both. Finally, deltas (or deviations) are obtained between the DVI and the DVG and between the PVI and the PVG. These comparisons give the measurement of value compliance, as well as the value losses present in the design. Figure 2 summarizes the model.

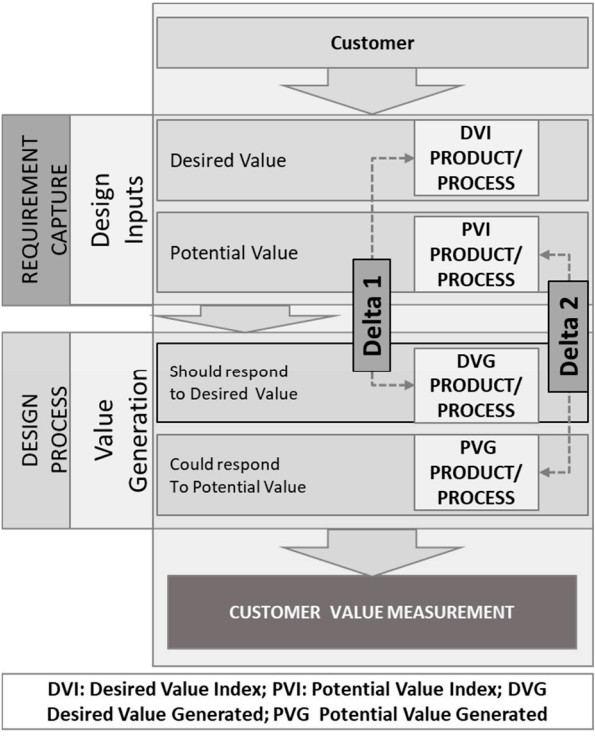

**Figure 2.** Value analysis model (VAM).

*3.2. Requirements Capture—Design Inputs*

3.2.1. Customer Identification

When starting to use the model, one customer must be identified, since value measurement must be done separately by customer or by customer groups by type: Designers, builders, owners, end-users, community members, etc. If it is necessary to review the value of different customers, the value measurement process must be repeated for each customer type.

3.2.2. Attribute List Creation

This list represents the standards to be evaluated by the customer. To make this list, the Delphi approach is recommended, in addition to the use of a literature review, a review of regulations and

standards, and previous experience. It is essential to consider the needs and requirements of the customer in the different ways in which they are incorporated without obtaining detailed specifications. Attributes should be clear, brief, and precise, and should avoid confusing or ambiguous terms. They should be formulated with a simple, direct, and familiar vocabulary for the participants, refer to only one aspect or logical relation, and be written positively [71]. If a list of attributes has already been created with another similar group, it can be validated with the new group or it can be started from the beginning, depending on the evaluation group's assessment of the value. The person answering the questions should understand that the default answers will reflect a classification, not a ranking. For this reason, the answers should not be misinterpreted as a rating on a scale of 1 to 5, so they should not be numbered [60].

### 3.2.3. Attribute Classification

The classification proposed by Kano et al. [18] is used. With the list of attributes, a two-dimensional questionnaire is prepared to assess each attribute. The first question is functional: How do customers feel if the proposed characteristic is provided? The second question is dysfunctional: How do customers feel if the intended characteristic is not provided? Each of the questions (whether functional or dysfunctional) has five response options: Like, must-be, neutral, live-with, and dislike. In this way, the attributes are classified by the customers themselves, to whom the questionnaire is administered. The attributes are then classified as M, O, R, A, and I attributes based on the matrix shown in Table 3. Q means that the question has probably been asked incorrectly or misinterpreted by the respondent.

**Table 3.** Kano's evaluation matrix.

| Functional | Dysfunctional | | | | |
|---|---|---|---|---|---|
| | Like | Must-be | Neutral | Live-with | Dislike |
| Like | Q | A | A | A | O |
| Must-be | R | I | I | I | M |
| Neutral | R | I | I | I | M |
| Live-with | R | I | I | I | M |
| Dislike | R | R | R | R | Q |

M = must-be, O = one-dimensional, R = reverse, A = attractive, I = indifferent, and Q = questionable.

Attribute classification generates a table with the list of attributes and the sums of the respondents' ratings. As an illustration, a fictitious example with 10 attributes is shown (see Table 4). Kano classifies each requirement according to most of the answers, which would not be statistically correct because, in general, the answers tend to be dispersed in several categories. In some cases, the first and second answers (even the third answer) are very close, and it is feasible to ask what the correct classification should be (see R3, R5, R7, and R9 in Table 4). For this reason, the CS proposed by Berger et al. [60] will be used.

Originally, in CS the R attributes were consciously ignored; the reason is not relevant and is beyond the scope of this paper. However, in this research, R attributes are included because it is important to determine which attributes a customer does not want to be present in the process or product. Considering that I attributes are neutral for the customer and that the inclusion of R is not desirable, it is preferable to classify an attribute as R instead of assuming that it is I (see R10 in Table 4). The R attributes will be included within the CS in the following manner:

$$SI = (O - R + A) / (M + O + R + A + I) \tag{3}$$

$$DI = (M + O + R) / (M + O + R + A + I) \tag{4}$$

*M*: Must-be, *O*: One-dimensional, *R*: Reverse, *A*: Attractive, *I*: Indifferent.

**Table 4.** Example of a classification table.

| Req | Must be | One-d | Reverse | Attract. | Indiffer. | Quest | Total | % 1st Response | Class. | Berger Original | | | Modified CS | | |
|-----|---------|-------|---------|----------|-----------|-------|-------|----------------|--------|-----|-----|-----|------|------|------|
| | **M** | **O** | **R** | **A** | **I** | **Q** | **T** | | **Kano** | **SI** | **DI** | **CS** | **SI-R** | **DI-R** | **CS-R** |
| R1 | 7 | 9 | 1 | 19 | 5 | 0 | 41 | 46% | A | 0.70 | 0.40 | A | 0.66 | 0.41 | A |
| R2 | 24 | 11 | 0 | 4 | 1 | 1 | 41 | 59% | M | 0.38 | 0.88 | M | 0.38 | 0.88 | M |
| R3 | 17 | 20 | 0 | 2 | 2 | 0 | 41 | 49% | O-M? | 0.54 | 0.90 | O | 0.54 | 0.90 | O |
| R4 | 3 | 8 | 2 | 6 | 21 | 1 | 41 | 51% | I | 0.37 | 0.29 | I | 0.30 | 0.33 | I |
| R5 | 19 | 17 | 0 | 4 | 1 | 0 | 41 | 46% | M-O? | 0.51 | 0.88 | O | 0.51 | 0.88 | O |
| R6 | 27 | 3 | 1 | 5 | 4 | 1 | 41 | 66% | M | 0.21 | 0.77 | M | 0.18 | 0.78 | M |
| R7 | 13 | 9 | 0 | 14 | 3 | 2 | 41 | 34% | A-M? | 0.59 | 0.56 | O | 0.59 | 0.56 | O |
| R8 | 29 | 4 | 0 | 7 | 1 | 0 | 41 | 71% | M | 0.27 | 0.80 | M | 0.27 | 0.80 | M |
| R9 | 0 | 0 | 19 | 0 | 22 | 0 | 41 | 54% | R-I? | 0.00 | 0.00 | I | −0.46 | 0.46 | I |
| R10 | 0 | 0 | 31 | 0 | 10 | 0 | 41 | 76% | R | 0.00 | 0.00 | I | −0.76 | 0.76 | R |

Berger et al. [60] initially established a graph with two axes between 0 and 1. By including the reverse attributes in the satisfaction index (SI) and the dissatisfaction index (DI), negative values are incorporated in the SI axis, as shown in Figure 3. A triangle incorporating values of M, I, and R is added to the initial four-quadrant graph to include the R attributes.

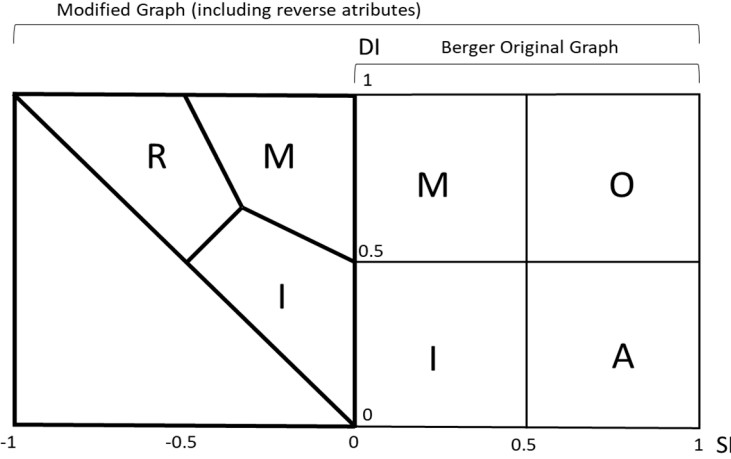

**Figure 3.** Modification of the Berger graph.

It is feasible that after using the CS, the values are in the limit between two types of attributes. If this happens, it is necessary to make a choice that must be made in the following order of priority: M > O/R > A > I. In other words, for example, if an attribute is on the boundary between A and I, it must be considered A.

### 3.2.4. Attribute Valuation

The attributes are related to value according to whether they are present or absent and their impact on customer satisfaction. A coding consisting of three values was applied: "−1" refers to customer dissatisfaction, "0" is neutral, and "+1" refers to customer satisfaction. Figure 4a shows the valuations proposed in VAM for each attribute based on the behavior graph of Kano's attributes. A attributes have a value of +1 if they are present and a value of 0 if they are absent. O attributes have a value of +1 if they are present and −1 if they are absent. If present, M attributes do not add value (0), but if absent, their value is negative. I attributes do not add value regardless of whether they are present or not. R attributes are valued positively if they are absent (+1) and negatively if they are present (−1). All valuations are summed in Figure 4b.

### 3.2.5. Calculation of Indexes

The DVI refers to what the customer expects. To calculate the DVI, only what is expected by the customer should be considered. Figure 4c shows the values expected by the customer for each type of attribute. A is not expected, so it is expected that it is absent, and its value would be 0; O and M are expected to be present; I does not matter if it is present or not; and R is expected to be absent. The DVI is the sum of the products of the number of type attributes and their valuation (in expected presence or absence) divided by the total attributes (Equation (5)). On the other hand, PVI refers to what the customer does not expect, it exceeds expectations. This model presents it as the sum of the DVI and percentage of A attributes (Equation (6)). Figure 5 illustrates the calculation of the indexes using the same types of attributes as in the example in Table 4.

$$DVI = (M * 0) + (O * 1) + (R * 1) + (A * 0) / M + O + R + A + I \qquad (5)$$

$$PVI = DVI + \%A \qquad (6)$$

*M*: Must-be, *O*: One-dimensional, *R*: Reverse, *A*: Attractive, *I*: Indifferent.

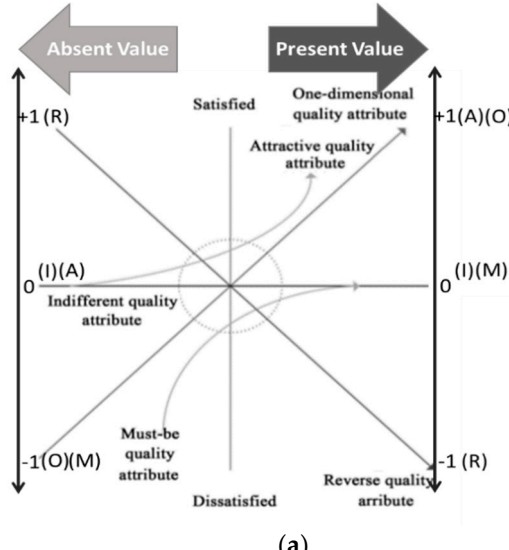

| Attributes | Value | |
|---|---|---|
| | Present | Absent |
| M | 0 | -1 |
| O | +1 | -1 |
| R | -1 | +1 |
| A | +1 | 0 |
| I | 0 | 0 |

(**b**)

| Attributes | Expected | |
|---|---|---|
| M | presence | 0 |
| O | presence | +1 |
| R | absence | +1 |
| A | absence | 0 |
| I | - | 0 |

(**c**)

**Figure 4.** Attribute valuation. (**a**) Behavior chart of Kano's attributes with the proposed valuations in the model; (**b**) present and absent value by type of attribute. (**c**) Expected attributes.

| Req | Quantity | Value | | Total | % |
|---|---|---|---|---|---|
| | | present | absent | | |
| M | 3 ✖ | 0 | → | 0 | 30% |
| O | 3 ✖ | +1 | → | 3 | 30% |
| R | 1 ✖ | | +1 → | 1 | 10% |
| A | 1 ✖ | | 0 → | 0 | 10% |
| I | 2 | - | - | 0 | 10% |
| | 10 | | | 4 | |
| | | | DVI | 4/10 | **0.40** |
| | | | PVI | | **0.50** |

**Figure 5.** Example of index calculation.

### 3.3. Design Process—Value Generation

When the design process formally begins, value begins to be generated. Therefore, the generated value indexes can be calculated and compared with the indexes calculated in the requirements capture stage.

### 3.3.1. Generated Value Indexes (GVI) Calculation

Based on the list of attributes already classified, designers will decide on the inclusion of the attributes requested by customers throughout the design process. For this measurement, a questionnaire with the same list of attributes and a percentage scale of presence was incorporated. The resultant values are used to quantify the level of presence and absence of each attribute type, and based on the valuations of each type of attribute, GVIs are calculated, as shown in Equations (7) and (8).

$$DVG = (Ma * -1) + (Op * 1) + (Oa * -1) + (Rp * -1) + (Ra * 1)/M + O + R + A + I \qquad (7)$$

$$PVG = (Ma * -1) + (Op * 1) + (Oa * -1) + (Rp * -1) + (Ra * 1) + (Ap * 1)/M + O + R + A + I \qquad (8)$$

*M*: Must-be, *O*: One-dimensional, *R*: Reverse, *A*: Attractive, *I*: Indifferent; suffixes *p* = level of presence and *a* = level of absence.

### 3.3.2. Comparison of Generated Value with Desired and Potential Value

Once the value generated is calculated, comparisons are made with the indexes established in the requirements capture. Figure 6 shows the relationships of requirements capture with value generation and identification of value losses and the relationship between the proposed value indices and the concept of value and attribute types. In order to obtain an initial value of "zero" as shown in the first bar, the M attributes must be fully met, since if they are not met or if they are not present, value is negative. Then, on this basis, the O attributes should be incorporated, and care should be taken to ensure that the R attributes remain absent to obtain the DVI. For the latter index, the A attributes are added to obtain the PVI. The I attributes do not add value.

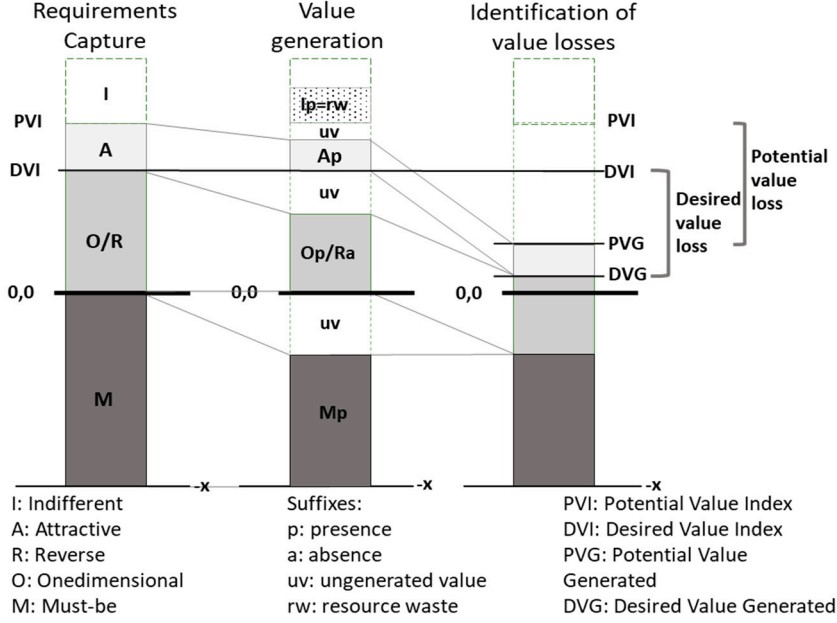

**Figure 6.** Relationships between the indexes, the attributes, and value losses.

Additionally, in the second bar, it is possible to observe the value generated in each of the types of attributes. The absence of M, O, and A represent ungenerated value, as well as the presence of R. On the other hand, when the I attributes are provided, they represent a waste of time, resources, and effort. Furthermore, two types of value losses are identified: (1) Those related to the desired value (such losses should be avoided completely); and (2) those related to the potential value (such losses could be avoided). Likewise, compliance percentages with value and loss of value of both the desired value and the potential have been incorporated to be used relatively and comparably. These key performance indicators (KPIs) are shown in Table 5 below.

**Table 5.** KPIs of the value generation process.

|  | Value Losses | Percentage of Value Losses | Percentage of Value Fulfillment |
|---|---|---|---|
| **Desired Value** | DVL = DVI-DVG | $DVLP = \frac{DVL}{DVI} \times 100$ | $DVFP = \frac{DVG}{DVI} \times 100$ |
| **Potential Value** | PVL = PVI-PVG | $PVLP = \frac{PVL}{PVI} \times 100$ | $PVFP = \frac{PVG}{PVI} \times 100$ |

| | | |
|---|---|---|
| DVI: Desired value index | | DVLP: Desired value loss percentage |
| PVI: Potential value index | DVL: Desired value loss | PVLP: Potential value loss percentage |
| DVG: Desired value generated | PVL: Potential value loss | DVFP: Desired value fulfillment percentage |
| PVG: Potential value generated | | PVFP: Potential value fulfillment percentage |

### 3.4. Value Evolution Over Time

#### 3.4.1. Determination of the Number of Revisions

The number of reviews of the value generated that will be made has to be established. These reviews can be incorporated in the project timeline frequently (weekly, fortnightly, monthly) or as milestones within the design process.

#### 3.4.2. Comparison with Other Reviews

Over time, the value generated within the design process can change and ideally should increase. With the different revisions, one could observe how PVG and DVG vary, as well as the losses in value. Figure 7 shows the different design iterations shown through different reviews. This graph is a simile of target costing, in which the target cost and the allowable cost are initially set, and the aim is to achieve them by reducing the costs through design decisions. Likewise, the PVI, i.e., the best possible value, and the DVI, i.e., the minimum value accepted by the customer, were fixed before starting the design process, and the iterations seek to reach PVI and DVI.

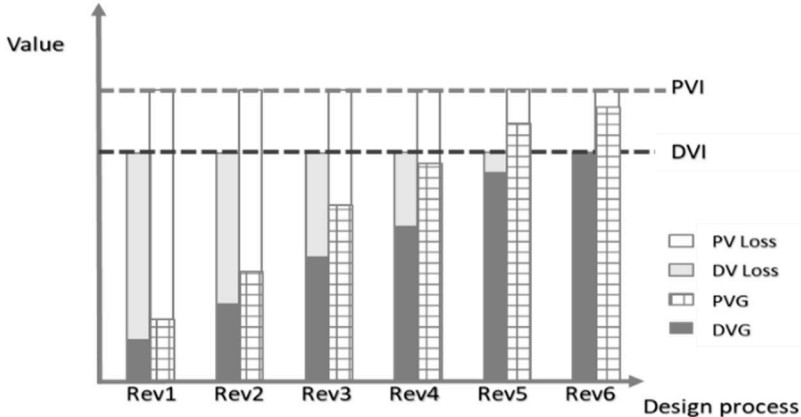

**Figure 7.** Value evolution over time.

### 3.5. Consideration of Multiple Customers

Design is an interactive and multidimensional effort that must represent the interests of several stakeholders [9]. In the context of this research, value is defined as the fulfillment of the needs of different customers or stakeholders, considering their diverse visions. For this reason, each customer can determine the desired value and potential value, and these values are probably very different from those of other customers. The considerations of several customers are shown in Figure 8.

Horizontally, as inputs for design, the requirements for the product or the design process of different customers are considered. In this sense, measurements result in total indexes of the product or process, both the desired and potential value of different customers. Likewise, throughout the design process, value is generated, which should respond to the requirements based on the desired value and could respond to the potential value.

For the calculation of these total indexes, a weighting factor (W) is established as the percentage value for each customer according to the importance given to the customer. The sum of all these Ws must be 1% or 100%, which will thus result in total indexes of desired value index (DVI), potential value index (PVI), desired value generated (DVG), and potential value generated (PVG) that amount to the total value of the product and of the process and that can be compared to each other.

Vertically, the total value is measured by identifying the deltas between the total desired value and the total potential value concerning the total value generated of both the process and the product. Ultimately, the overall result of the model will be the total measurement of value concerning the whole design process and considering all customers.

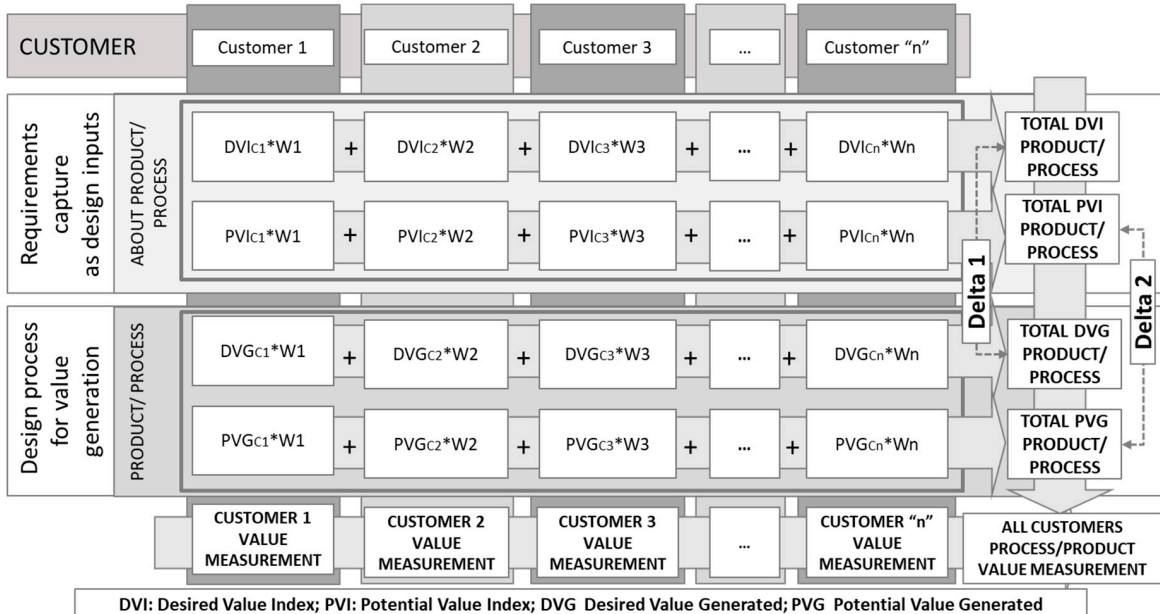

**Figure 8.** Complete VAM considering multiple customers.

## 4. Analysis of a Practical Implementation

The model was implemented in the first design stages of two projects of a real estate and construction company located in Santiago de Chile, whose primary activity is the integral execution of high-rise residential buildings. Project 1 and Project 2 were selected as case studies for their similar characteristics of scope, user profiles, and level of design progress, in addition to the researcher's access to the stakeholders involved.

VAM was applied to three customers (owner, designer, and builder) in three different aspects: Design process, product, and by-products in only one review. For this particular paper, the interest is to show how the practical application of VAM was performed, which is why only a part of the practical test will be shown below. The results that are shown refer specifically to how VAM works and what it can achieve.

The product value measurement results are shown in Tables 6 and 7. Table 6 shows the attributes list, and on the Table 7 illustrates the summary of the results of the product value measurement of the three customers in both projects. First, the perceptions of value of each customer differ; a higher DVI indicates that a customer expects more value than another with a lower DVI. However, the effort to meet this customer's expectations depends not only on the DVI but also on the percentage of M attributes as the basis. A DVI close to "zero" represents many M and/or I attributes, and a DVI close to "1" requires many O and R attributes (to be avoided), well above the number of M, I, and A attributes. On the other hand, a PVI with values close to the DVI means that few A attributes are identified, which can induce the need to innovate to include this type of attribute in the list.

Second, even when these results were not expected, negative values were observed in the GVIs and, therefore, in the fulfillment percentages of desired or potential value, which means that the value loss is very high (more than 100%). In this case, the percentages of value losses incorporated facilitate understanding when negative value is generated.

In all cases, PVG is higher than DVG, which could be natural. However, this result means that even if the desired value has not been met in its entirety (M + O and avoiding R), efforts are being made to achieve A attributes, which shows that there is no clear prioritization of tasks.

This information confirms that the desired value of customers is not being generated. The next step consists of reviewing the compliance percentage of each type of attribute to determine which aspects of value are lost or generated. Table 8 illustrates in detail the generation of value in the design

process for the designers in Project 2. The compliance percentages of attributes M and O are below those of attributes A and the same for attributes I, the latter representing a waste of resources and efforts in the compliance of attributes that do not generate value for the customer and confusing prioritization in the alignment of design objectives.

**Table 6.** Attributes list of product.

| | Attributes List |
|---|---|
| 1 | High percentage of repetitive elements |
| 2 | Low cost variability |
| 3 | Good cost/quality ratio |
| 4 | Good value for money/square meters |
| 5 | Good location |
| 6 | Sellable/competitive design |
| 7 | Aesthetic |
| 8 | Easy to build |
| 9 | Functional |
| 10 | Differentiating image |
| 11 | Innovative |
| 12 | Materials available on the market |
| 13 | Durable materials |
| 14 | Easy to install materials |
| 15 | Product stable to earthquakes and other events |
| 16 | Profitable product |
| 17 | Compliant with regulations |
| 18 | That meets the customer's requirements |
| 19 | To maintain its value over time |
| 20 | To improve the quality of life of the community |
| 21 | To improve the customer's quality of life |
| 22 | No reclaims |
| 23 | Presenting cutting-edge technology |
| 24 | Sustainable/energy efficient |

**Table 7.** Product value measurement.

| | Customer | | | Total |
|---|---|---|---|---|
| | **Owner** | **Designers** | **Builders** | |
| w | 42% | 28% | 31% | 100% |
| DVI | 0.79 | 0.32 | 0.29 | 0.51 |
| PVI | 1.00 | 0.69 | 0.42 | 0.73 |
| **Project 1** | | | | |
| DVG | 0.10 | −0.11 | −0.11 | −0.02 |
| PVG | 0.22 | 0.14 | −0.04 | 0.12 |
| DVFP | 13% | −34% | −38% | −4% |
| PVFP | 22% | 20% | −10% | 16% |
| DVL | 0.69 | 0.42 | 0.40 | 0.53 |
| PVL | 0.78 | 0.55 | 0.46 | 0.62 |
| DVLP | 87% | 134% | 138% | 104% |
| PVLP | 78% | 80% | 110% | 84% |
| **Project 2** | | | | |
| DVG | 0.39 | 0.07 | 0.07 | 0.20 |
| PVG | 0.54 | 0.34 | 0.16 | 0.37 |
| DVFP | 49% | 23% | 24% | 40% |
| PVFP | 54% | 49% | 39% | 50% |
| DVL | 0.40 | 0.24 | 0.22 | 0.30 |
| PVL | 0.46 | 0.35 | 0.26 | 0.37 |
| DVLP | 51% | 77% | 76% | 60% |
| PVLP | 46% | 51% | 61% | 50% |

**Table 8.** Value generation in Project 2—process-designers.

| | Process—Project 2 | | | |
|---|---|---|---|---|
| | Designer | | | |
| | % Present | V. Pre | V. Abs | Score |
| M | 69% | 0 | −1 | −2.80 |
| O | 69% | 1 | −1 | 2.27 |
| A | 80% | 1 | 0 | 3.99 |
| I | 69% | 0 | 0 | 0.00 |
| R | | −1 | 1 | 0.00 |
| | DVL | 0.27 | DVG | −0.02 |
| | PVL | 0.31 | PVG | 0.14 |
| | DVLP | 109% | DVFP | −9% |
| | PVLP | 69% | PVFP | 31% |

It is also possible to establish comparisons between projects. In this case, two projects with similar characteristics within the same company are compared, which is why there is only one DVI and PVI by customer. Projects with a different DVI and PVI for the customer are possible. However, comparisons concerning the relative value generated and value loss can be made. Notably, Project 2 has created higher value than Project 1 (see Table 7), but it still has value losses that must be covered.

## 5. Conclusions

### 5.1. Summary

This paper identifies a gap in current practices in the value generation process in design within the AEC industry: There is a lack of adequate methods that link the suitable capture of customer requirements with the continuous measurement of the value generated as well as the timely identification of value losses at the time of design and not later, when it is no longer feasible to deal with them. A model is proposed to measure the value creation expected by customers and to identify value losses through indexes, which can help designers and project managers improve decision making within the design process, increase customer satisfaction and evaluate the allocation of resources to those activities that actually generate value.

### 5.2. Contributions

The proposed model responds to the need to measure the value creation expected by different customers within the design process through indexes of desired, potential, and generated value and the percentages of the fulfillment of desired and potential value. In addition, the model connects with the concept of value losses [3] and contributes to the numerical and graphical identification of such losses. Likewise, it is capable of showing to interested individuals the aspects in which value is generated and other aspects in which it is partially or completely lost. The model supports a better understanding of the concept of value and how to capture it to support the conditions of customer satisfaction.

The VAM enables an integral view of the whole process encompassing the total measurement, considering the process, product, and customers. This vision can be incorporated for a particular aspect. Moreover, the percentage of incorporation of each type of attribute in design decisions provides clarity on the issues to which more significant efforts and resources should be allocated (to incorporate M and O attributes and to avoid R attributes), and the other aspects to which moderate efforts and resources should be allocated (incorporation of I attributes). Additionally, different comparisons can be made between different value visions of customers, the differences between the value generated by the process and product, the differences between the value generated in several projects, the differences between the value generated per customer, etc. In the same terms, it is possible to compare value losses

per customer, per project, or between the process and product. On the other hand, it is possible to see the evolution of the value generated over time with several revisions.

The proposed model possesses certain flexibility and adaptability for diverse research needs. It can be applied in a specific area, for example, if there is a desire to evaluate what value is generated in terms of sustainability or security conditions or to choose which elements are the most attractive to the customer concerning the common areas of a building. Similarly, the value expectations of one target population can be compared to another, or how different design schemes or methodologies meet customer satisfaction conditions can be evaluated. Likewise, the evolution of the different indexes over time can be studied to reveal dynamic changes in customer preferences.

The development of the VAM contributes to knowledge since it responds to the challenge of defining and generating value in the design process, taking into consideration customers' requirements as process inputs. In addition, the VAM is based on influential factors for the generation of value and can show the impact of decisions or the use of methodologies on value generation or loss.

The model has practical value within the AEC industry. It is useful for optimizing products and processes since aspects for continuous improvement of the process are identified promptly by stages and by projects. It encourages constant feedback and has the potential to provide a higher delivery of value, as it makes it possible to determine the parameters that add value for different stakeholders, thereby informing designers where to direct resources and efforts to enhance vital variables and not trivial variables. In the VAM practical implementation, the design team considered the requirements of the builders in detail to improve the constructability and standardization of both projects, as well as the replacement of some elements and materials to make them optimal.

The VAM allows the observation of changes in value over time and how these changes align with the decisions made. Additionally, the model encourages conversations among key actors, makes it possible to think about value for the next customer in the process, and constitutes a contribution to adequately capturing requirements. In practical implementation, the professionals consulted considered VAM as a good tool for collaborative development, since it makes information and communication between the different stakeholders transparent, achieving clear requests from the early stages. A correct future implementation helps to have a differentiating element compared to other companies.

### 5.3. Limitations

The practical testing focused on two vertical building projects and was based on the experience of 20 professionals in building construction and design. Therefore, the results should not be interpreted as universal to all types of construction projects. However, the VAM is believed to be applicable to other sectors, such as housing, industrial construction, and infrastructure. In addition, when the model was tested, no consideration was given to the perception of value of the end user or the use of partial or total resources in increasing the value in the project.

### 5.4. Future Research

Opportunities for future research include the VA of other stakeholders, mainly end-users, as well as VA in other sectors of the AEC industry, not only vertical housing building. The possibility of continuously measuring value will be addressed in a further paper, which will incorporate not only different steps of the design process, but also other customers. The ability to capture the value perspectives of different stakeholders is a beneficial aspect of the VAM. However, these stakeholders are expected to present conflicting requirements, as their interests may be very different from each other; thus, the model can provide recommendations on how to weigh stakeholder requirements in the event of incompatibilities. It may be appropriate to include stakeholder mapping as support.

Concerning the resources used in the design process, the model shows how much effort and resources have generally been allocated to unimportant aspects, such as compliance with I attributes.

The cost variable or the evaluation of the use of reallocation costs from less desirable to more desirable attributes could be added as a parallel axis.

**Author Contributions:** Conceptualization, Z.G.; formal analysis, Z.G.; investigation, Z.G.; methodology, all authors; development and implementation of the model, Z.G.; writing—original draft, Z.G.; writing—review and editing, C.M., L.F.A., H.M., and E.P.; supervision, C.M., L.F.A., and E.P. All authors have read and agreed to the published version of the manuscript.

**Funding:** This research was funded by CONICYT grant number PCHA/National Doctorate/2016-21160571 for the postgraduate studies of Zulay Giménez and by FONDECYT (1181648).

**Acknowledgments:** The authors wish to thank the organization participating in this study as well as the experts for the insight provided.

**Conflicts of Interest:** The authors declare no conflict of interest.

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
