# Peer review of "Value Analysis Model to Support the Building Design Process"

_sustainability, doi:10.3390/su12104224_

Round 1

Reviewer 1 Report

Dear Authors,

I found your research study very interesting. You described very well the VAM methodology, so I have no comment regarding this.

You have mentioned that VAM can be used for the continuous measurement of the value generated and identification of value losses. Have you tested this methodology in different steps of the design process? Using this methodology, were your customers/stakeholders able to make corrective decision to improve and deliver the desired value? It would be interesting to understand if and how this methodology allowed your customers and stakeholders to improve the design process. Also, it would be interesting to know what was the customer/stakeholders feedback in the practical implementation. Were they satisfied? 

Could you include your customer/stakeholder feedback regarding VAM in the practical implementation section or conclusions?

Best regards.

Author Response

The authors thank Reviewer #1 for the time and effort in reviewing our manuscript and the suggestions, comments, and observations. The following point-by-point response explains the changes in the document, following the recommendations. We believe that the paper is much improved.

Reviewer 2 Report

GENERAL COMMENTS

The purpose of this paper is to respond to the need to measure the value creation expected by different customers within the design process through indicators and to contribute to the early identification of value losses to control them in time. A value analysis model is proposed by the authors to evaluate the value creation expected by customers and to identify value losses in the building project design process through indexes based on the Kano model and target costing as points of reference. 

The proposed model responds to the need to measure the value creation expected by different customers within the design process through indexes of desired, potential, and generated value and the percentages of the fulfillment of desired and potential value. In addition, the model connects with concept of value losses and contributes to the numerical and graphical identification of such losses.

The paper is very interesting and presents important practical implications for the AEC industry. Moreover, it is clear and well structured.

Author Response

The authors thank Reviewer #2 for the time and effort in reviewing our manuscript and the very positive and encouraging comments.

Reviewer 3 Report

The authors develped based on literature review and Extended model for the execution and Evaluation of a buliding desing process. The newly developed value Analysis models contains a Synopsis of severals different aspects and perspectives of Stakeholders.  Determination and Integration of customer value into the design process.  The model has been applied/verified in practical implementations. The method allows to tranform "opinions", "Feelings" of various stakeholders into measuralbe values for transparent decisions About the optimal design. The model is adaptable and very felxible cocerrning modifications and the Integration of different Parameters and Stakeholders.

As the authors also state one weak Point in the entire model is the "perception of value of the end user".  This is in my oppinion an essential criterion because in a lot of cases the end users might als obe the opne who Pays for the Building and/or the usage. I wonder why These criteria couldn´t be included in the model and "weighted" accordingly. A weightening System would also solve or at least smoothening the adressed Problem of conflicting requirments. 

Author Response

The authors thank Reviewer #3 for the time and effort in reviewing our manuscript and the suggestions, comments, and observations. The following point-by-point response explains the changes in the manuscript, following the suggestions. We believe that the paper is much improved.

Reviewer 4 Report

Congratulations - very interesting subject. Very good is to use DSR. It is logical to take into consideration Kano model. Table 2 is really confidential (maybe someone could be identified), please? There are two Deltas in Figure 2, which measures different value - maybe it is better to give them different names?
It is very good idea to comparison of different projects. In my opinion the majority of non-quality is generated by design (in the construction industry).
Good luck for more case studies and implementation in construction practice.

Author Response

The authors thank Reviewer #4 for the time and effort in reviewing our manuscript and the suggestions, comments, and observations. The following point-by-point response explains the changes in the manuscript, following the suggestions. We believe that the paper is much improved.
